SARS-CoV-2-specific humoral and cellular immunity assessment in Peruvian vaccinated population: a cross-sectional study

http://orcid.org/0000-0003-3028-3454 Garcia-Paitan Marlon Yuri 1 2 3
Tapia-Rojas Salyoc 1 2
Alvarez Vega Hector Santiago 1 2
Enciso-Benavides Javier 4
http://orcid.org/0000-0001-8384-2315 Pons Maria J. 4
Mayanga-Herrera Ana 1 2 amayanga@cientifica.edu.pe
1 Cell Culture and Immunology Lab, Universidad Cientifica del Sur , Lima , Peru
2 Cancer and Stem Cells Research Group, Universidad Cientifica del Sur , Lima , Peru
3 Unidad de Posgrado de la Facultad de Ciencias Biológicas, Universidad Nacional Mayor de San Marcos , Lima , Peru
4 Grupo de Enfermedades Infecciosas Re-Emergentes, Universidad Cientifica del Sur , Lima, Lima , Peru
Hromić-Jahjefendić Altijana
Electronic publication date: 2025 Jul 15
Publication date: 2025
Volume: 13
Electronic Location ID: e19651
Received 2025 Feb 17; Accepted 2025 Jun 2
Copyright: © 2025 Garcia-Paitan et al.
Copyright year: 2025
Copyright holder: Garcia-Paitan et al.
License: This is an open access article distributed under the terms of the Creative Commons Attribution License, which permits unrestricted use, distribution, reproduction and adaptation in any medium and for any purpose provided that it is properly attributed. For attribution, the original author(s), title, publication source (PeerJ) and either DOI or URL of the article must be cited.
License URL: https://creativecommons.org/licenses/by/4.0/

Keywords: Cellular immunity, SARS-CoV-2, Vaccine, RT-qPCR, Humoral immunity, CXCL10 mRNA

Funding: PROCIENCIA-AECID PE501079519-2022 Universidad Científica del Sur 062-2023-PRO99 This study has been funded by PROCIENCIA-AECID (PE501079519-2022) and the Universidad Científica del Sur (062-2023-PRO99). The funders had no role in study design, data collection and analysis, decision to publish, or preparation of the manuscript.

==============================
Background

Evaluating both humoral and cellular immunity is essential for optimizing vaccination strategies and preventing post-pandemic SARS-CoV-2 outbreaks. This cross-sectional study assessed cellular immunity by measuring CXCL10 mRNA expression and humoral immunity through SARS-CoV-2-specific IgG antibodies.

Method

Whole blood samples were collected from 40 Peruvian volunteers. CXCL10 expression was evaluated in blood samples stimulated with Spike protein peptides from the Wuhan strain and Omicron BA.5 variant using RT-qPCR. Anti-spike IgG levels were measured with a semi-quantitative ELISA.

Results

The median age was 31 years, with 62.5% females. A heterologous vaccination scheme was reported by 73%, but only 25% received their last dose within the past 6 months, and 55% completed three doses. The BNT162b2 vaccine was included in 88% of vaccination schemes, serving as the first and second dose in 48% of cases. All participants had detectable anti-spike IgG antibodies; 90% exhibited cellular responses to Wuhan peptides and 97.5% to Omicron peptides. CXCL10 mRNA expression (2−ΔΔCT) was significantly higher for Omicron (median: 565.97; IQR: 565,148.34) compared to Wuhan (median: 18.55; IQR: 62,898.67). Higher anti-spike IgG levels correlated with age and the number of vaccine doses. Males had significantly higher CXCL10 and anti-spike IgG levels (p < 0.05). Antibody levels were greater in those recently boosted or vaccinated with mRNA-1273 (p = 0.001, p = 0.002).

Conclusion

Most participants exhibited robust immunity, characterized by elevated levels of CXCL10 and anti-SARS-CoV-2 IgG antibodies. These findings highlight the importance of boosters in enhancing immunity and the need for diverse techniques for measuring immunity.

Introduction

The COVID-19 pandemic has posed unprecedented challenges to public health systems, particularly due to the emergence of variants of concern (VOCs). These variants have triggered multiple waves of infections, complicating strategies to control viral spread and undermining vaccine effectiveness (Rabaan et al., 2023). Such challenges arise from their immune evasion capacity, high transmissibility, increased virulence, and elevated infection rates (Choi & Smith, 2021).

In the post-pandemic phase, evaluating population-level protective immunity is crucial to provide up-to-date data on vaccine effectiveness, particularly with extended intervals between doses, and assessing variability based on demographic factors, vaccination schedules, and prior SARS-CoV-2 infection (Schwarz et al., 2022a; Alroqi et al., 2024). Variability in vaccine efficacy has been documented in vaccines for diseases such as malaria, yellow fever, and rotavirus across populations due to region-specific environmental factors (Van Dorst et al., 2024). Thus, immunological studies on new vaccines in diverse regions are essential. Accurate vaccine effectiveness data are critical to design or adjust vaccination programs and control measures tailored to population-specific characteristics, aiming to prevent future infection waves from emerging or re-emerging VOCs (Telenti et al., 2021; Schwarz et al., 2022a; Hall et al., 2022; Malani et al., 2024).

Evaluating protective immunity against SARS-CoV-2 requires a comprehensive understanding of both the humoral and cellular arms of the adaptive immune response. While antibodies play a central role in preventing initial infections, T cells are vital for controlling infection progression (Schwarz et al., 2022a). Regarding functional durability, neutralizing antibody titers decline more rapidly than T cell memory, which persists for longer periods (Almendro-Vázquez, Laguna-Goya & Paz-Artal, 2023). Moreover, T cell responses are more robust against VOCs, capable of cross-recognizing highly mutated variants such as BA.2.86 and maintaining reactivity against Omicron sub lineages, contrasting with the limited efficacy of neutralizing antibodies (Nesamari et al., 2024).

Assessing cellular immunity has faced technical challenges that have limited its use in population-based studies. To address this, Schwarz et al. (2022b) identified six genes whose expression positively correlates with interferon γ (IFN-γ), a key marker of T cell activation, following stimulation of whole blood with Spike-Gold peptides. Among these genes, CXCL10 emerged as the most reliable and reproducible biomarker in qPCR assays due to its low variance. Consequently, measuring CXCL10 expression after ex vivo stimulation with SARS-CoV-2 peptides offers an accessible tool for evaluating cellular immunity against this virus in the post-pandemic context.

In Peru, the SARS-CoV-2 vaccination program began in early 2021 with a homologous two-dose regimen using BBIBP-CorV (Sinopharm), AZD1222 (Oxford-AstraZeneca), and BNT162b2 (Pfizer-BioNTech) vaccines (López et al., 2022). By late 2021, booster doses were introduced, employing both homologous and heterologous schemes, including the mRNA-1273 (Moderna) vaccine, with a minimum interval of 3 months after the second dose (García-Mendoza et al., 2022). In 2024, monovalent adapted vaccines (Comirnaty Omicron XBB.1.5) and bivalent vaccines (Spikevax Bivalent Original/Omicron BA.4-5) were approved. To date, national coverage has reached 94.17% with at least one dose, 90.48% with two doses, 74.88% with three doses, and 28.61% with four doses (Ministry of Healthy of Peru (MINSA), 2024).

However, studies on SARS-CoV-2 immunity in Peru have focused primarily on humoral immunity in healthcare workers, with no data available on cellular immunity in the general population. This gap is significant, particularly as booster dose administration has markedly declined. Although the infection is now considered endemic, inadequate surveillance could facilitate the emergence of new epidemic or pandemic patterns (Telenti et al., 2021). Therefore, this study aims to evaluate humoral immunity through anti-spike IgG antibody detection and cellular immunity via CXCL10 mRNA expression in response to the Omicron variant and the original Wuhan strain in vaccinated individuals in Peru.

Materials and Methods

Study design

A cross-sectional study was conducted between August and September 2023, involving volunteers from the Peruvian population who had been vaccinated against SARS-CoV-2. Participants were recruited at the Universidad Científica del Sur in Lima, Peru. After providing signed written informed consent, participants completed a clinical-epidemiological questionnaire, and a blood sample was drawn from the antecubital vein for further analysis.

Research ethics

The study was conducted in accordance with the Declaration of Helsinki and was approved on July 13, 2023, by the Institutional Review Board of Universidad Científica del Sur (CIEI-CIENTIFICA) under approval number 173-CIEI-CIENTIFICA-2023.

Blood sample collection

A total of 6 mL of peripheral blood was collected from the antecubital vein into vacuum tubes containing heparin as an anticoagulant. Immediately after collection, the samples were gently inverted 5–10 times to ensure mixing with the anticoagulant and processed within 6 h for the stimulation assay. One mL of blood was centrifuged, and the supernatant was collected to obtain plasma for ELISA assays.

Measurement of anti-S1 IgG antibodies

A commercial Anti-SARS-CoV-2 ELISA kit (EuroImmun, Lübeck, Germany) was utilized to evaluate anti-Spike IgG antibodies against SARS-CoV-2, following the manufacturer’s instructions. Serum samples were diluted 1:101 in sample buffer, and 100 µL of the diluted sample, calibrator, and controls were added to the respective wells. After a 60-min incubation at 37 °C, the plates were washed three times and incubated for 30 min at 37 °C with an enzyme-conjugated anti-human IgG (peroxidase-labeled). Subsequently, 100 µL of TMB substrate solution was added and incubated for 30 min at room temperature. The reaction was stopped with 0.5 M sulfuric acid, and optical density (OD) was measured at 450 nm with a 620–650 nm reference using a Biotek Synergy LX Multi-Mode Reader (Agilent, Santa Clara, CA, USA). Results were interpreted semiquantitatively by calculating the ratio of the OD of the sample to that of the calibrator. Following manufacturer recommendations, results were interpreted using the following ratio thresholds: negative (<0.8), borderline (≥0.8 to <1.1), or positive (≥1.1). For statistical analysis, we applied a base-10 logarithmic transformation to the ratio values, yielding corresponding transformed thresholds: negative (log10[ratio] < −0.097), borderline (log10[ratio] ≥ −0.097 to <0.041), and positive (log10[ratio] ≥ 0.041).

SARS-CoV-2 peptides stimulation

The stimulation assays were conducted in 96-well plates, utilizing 1 mL of whole blood from each participant, 250 uL were distributed in four wells corresponding to the experimental groups: Negative Control, Positive Control, Wuhan Stimulation, and Omicron Stimulation. The Negative Control group consisted of whole blood mixed with a 20% DMSO solution (peptide diluent) in a volume equivalent to that used in the stimulation groups. In the Positive Control group, phytohemagglutinin was added at a final concentration of 10 μg/mL. The Wuhan Stimulation group was treated with a final concentration of 0.5 μg/mL of PepTivator® SARS-CoV-2 Prot_S (Miltenyi Biotec, Santa Barbara, CA, USA), consisting predominantly of 15-mer sequences with an 11-amino-acid overlap, spanning the Spike glycoprotein of the SARS-CoV-2 Wuhan wild-type strain. Similarly, the Omicron Stimulation group was treated with 0.5 μg/mL of PepTivator® SARS-CoV-2 Prot_S Complete BA.5 (Miltenyi Biotec, Santa Barbara, CA, USA), consisting of 15-mer peptides with an 11-amino-acid overlap corresponding to the complete Spike glycoprotein of the Omicron variant (lineage BA.5). All samples were incubated at 37 °C with 5% CO2 for 18 h.

RNA extraction

Total RNA was extracted from each experimental and control group using the Trizol method, with modifications based on the protocol described by Gautam et al. (2019). Briefly, 200 µL of each stimulated sample was combined with 600 µL of Trizol, vortexed briefly, and stored at −80 °C. Prior to processing, samples were thawed at room temperature and an additional 400 µL of Trizol was added to the mixture, followed by vortexing for 30 s. Subsequently, 200 μL of chloroform was added, vortexed for 30 s, and centrifuged at 12,000× g for 15 min at 4 °C. The upper aqueous phase, 600 μL, was carefully transferred to a clean 1.5 mL tube and mixed with an equal volume of cold isopropanol. After inversion mixing, the mixture was incubated at −20 °C for 10 min and centrifuged at 12,000× g for 15 min at 4 °C. The resulting pellet was washed three times with 1 mL of 75% ethanol, vortexed gently, and centrifuged at 7,500× g for 5 min at 4 °C. After the final wash, the pellet was air-dried at room temperature for 10 min, resuspended in 25 µL of RNase-free water, and stored at −80 °C until further use.

One-step real-time PCR amplification of CXCL10

The protocol described by Schwarz et al. (2022b) was followed for the qualitative assessment of CXCL10 gene expression, with ACTB used as an internal control. The following primers and probe were used for CXCL10 detection: forward primer (CCATTCTGATTTGCTGCCTTATC), reverse primer (TACTAATGCTGATGCAGGTACAG), and probe (FAM-AGTGGCATTCAAGGAGTACCTCTCTCT-BHQ-1). For ACTB amplification, the primers and probe used were: forward primer (CCTTGCACATGCCGGAG), reverse primer (ACAGAGCCTCGCCTTTG), and probe (ROX-TCATCCATGGTGAGCTGGCGG-BHQ-2). The primers and probes in both cases are designed to target exon–exon junctions. Reverse transcription quantitative PCR (RT-qPCR) was performed using the SCRIPT Direct RT-qPCR ProbesMaster kit (Jena Bioscience, Jena, Germany), with reactions prepared according to the manufacturer’s instructions. The PCR amplification was carried out in a LineGene K thermocycler model FQD-48A (BIOER, Hangzhou, China) as follows: an initial reverse transcription step at 50 °C for 25 min, followed by an initial denaturation at 95 °C for 5 min. This was followed by 45 amplification cycles, each consisting of denaturation at 95 °C for 15 s and extension at 60 °C for 30 s, with fluorescence captured during the extension phase. Relative gene expression was analyzed using the 2−ΔΔCT method (Livak & Schmittgen, 2001), with data processing conducted using LineGene4800 software, which was also used to control the PCR program on the LineGene K thermocycler.

Statistical analysis

A descriptive analysis was conducted to summarize the demographic and clinical characteristics of the study subjects. For categorical variables, relative frequencies were calculated, while for continuous variables, the mean, median and interquartile range (IQR) were reported. Graphical plots and statistical analyses (Student’s t-test, Mann-Whitney U test, Pearson and Spearman correlation coefficient, depending on the nature of the variables.) were performed using RStudio (Version 1.4.1717; RStudio Team, 2021). Multivariate analyses, including linear regression, were also conducted. Analyses were based on the −ΔΔCt values (log2 of 2−ΔΔCt) for CXCL10 RT-qPCR in the Omicron and Wuhan stimulation groups, as well as the log10 of the ratio for IgG anti-spike antibodies against SARS-CoV-2.

Results

Study population

A total of 40 samples were included in the study. The participants had a mean age of 31 years, with 62.5% being female. Of the participants, 38% reported a previous SARS-CoV-2 infection prior to receiving their first dose of the COVID-19 vaccine. Among the 36 participants with a history of SARS-CoV-2 infection, 32 (90%) experienced mild to moderate symptoms, three (8.3%) had severe cases, and one (2.8%) was asymptomatic. Fifty-five percent of participants reported receiving three doses of the COVID-19 vaccine, and 29 (73%) followed a heterologous vaccination regimen, with BNT162B2 being the most commonly administered vaccine (88%). Furthermore, 45% and 48% of participants received BBIBP-CorV and BNT162B2, respectively, as their first two doses. Twenty-five percent of participants received their most recent dose of the COVID-19 vaccine within the last 6 months (Table 1).

Table 1 Characteristics and vaccination history of study participants by sex.

Characteristic	Overall	Female	Male	p-value	
N = 401	N = 251	N = 151	
Age	31 (25, 40)	30 (24, 37)	32 (26, 50)	0.282	
Chronic disease	14/40 (35%)	9/25 (36%)	5/15 (33%)	0.864	
Immunosuppression history	2/40 (5.0%)	1/25 (4.0%)	1/15 (6.7%)	1.000	
Fever episode (Last 3 months)	8/40 (20%)	7/25 (28%)	1/15 (6.7%)	0.219	
SARS-CoV-2 infection (previous to sample collection)	36/40 (90%)	22/25 (88%)	14/15 (93%)	1.000	
SARS-CoV-2 infection (previous to first vaccination)	15/39 (38%)	9/24 (38%)	6/15 (40%)	0.876	
ARNm-1273	12/40 (30%)	7/25 (28%)	5/15 (33%)	0.736	
AZD1222	6/40 (15%)	4/25 (16%)	2/15 (13%)	1.000	
BBIBP-CorV	20/40 (50%)	13/25 (52%)	7/15 (47%)	0.744	
BNT162b2	35/40 (88%)	21/25 (84%)	14/15 (93%)	0.633	
Number of doses				0.318	
2	1/40 (2.5%)	0/25 (0%)	1/15 (6.7%)		
3	22/40 (55%)	16/25 (64%)	6/15 (40%)		
4	11/40 (28%)	6/25 (24%)	5/15 (33%)		
5	6/40 (15%)	3/25 (12%)	3/15 (20%)		
First 2 doses (Same vaccine type)				0.677	
AZD1222	1/40 (2.5%)	0/25 (0%)	1/15 (6.7%)		
BBIBP-CorV	18/40 (45%)	12/25 (48%)	6/15 (40%)		
BNT162b2	19/40 (48%)	12/25 (48%)	7/15 (47%)		
Criteria not met	2/40 (5.0%)	1/25 (4.0%)	1/15 (6.7%)		
Vaccine scheme				0.486	
Heterologous	29/40 (73%)	17/25 (68%)	12/15 (80%)		
Homologous	11/40 (28%)	8/25 (32%)	3/15 (20%)		
Time since last dose				0.457	
>6 Months	30/40 (75%)	20/25 (80%)	10/15 (67%)		
≤6 Months	10/40 (25%)	5/25 (20%)	5/15 (33%)		
Notes:

Vaccines: ARNm-1273 (Moderna), AZD1222 (AstraZeneca), BBIBP-CorV (Sinopharm) and BNT162b2 (Pfizer).

1 Median (Q1, Q3); n/N (%).

2 Wilcoxon rank sum test; Pearson’s Chi-squared test; Fisher’s exact test.

Cellular response by measuring CXCL10

A 260/280 absorbance ratio of 1.88 ± 0.16 was obtained following the RNA extraction protocol. A subset of samples was tested without reverse transcription using two-step RT-qPCR. No amplification was observed, confirming the absence of genomic DNA contamination and ensuring primer and probe specificity. Stimulation assays using SARS-CoV-2 peptides were validated in all participants, as the relative quantification of CXCL10 expression in the positive control (Phytohemagglutinin-stimulated) exceeded that of the negative control. One case showed no amplification of CXCL10 in the negative control; therefore, a correction factor was applied based on the ΔCt values from other negative controls with detectable amplification (Cosma et al., 2023).

Positivity of cellular immunity to SARS-CoV-2 peptides was assessed according to the methodology outlined in quantitative PCR rapid T-cell activation assay (Schwarz et al., 2022b). Positive cellular responses to Omicron variant peptides were observed in 39 participants (97.5%), with only one negative response (2.5%) and a median 2−ΔΔCT value of 565.97 (IQR: 565,148.34).

For the original Wuhan strain peptides, positive responses were detected in 36 participants (90%), with four negative responses (10%). The median 2−ΔΔCT value for Wuhan peptides was 18.55 (IQR: 62,898.67).

These results demonstrated a robust cellular immune response in most participants, with a higher response rate and median CXCL10 expression for Omicron peptides compared to those from the original Wuhan strain (Wilcoxon test, p < 0.001).

Immunity levels in relation to categorical clinical variables

Figure 1 illustrates the relative expression levels of CXCL10 and IgG anti-spike antibody titers according to sex, time since the last vaccination, and vaccination regimen. Significant differences were observed between males and females, with higher levels of CXCL10 expression and IgG anti-spike antibodies in the male population (median −ΔΔCt Wuhan: 7.70 vs 2.50, Wilcoxon test, p = 0.007; mean −ΔΔCt Omicron: 15.44 vs 7.62, t-test, p = 0.001; median Log10 IgG anti-spike: 1.107 vs 1.024, Wilcoxon test, p = 0.033).

Figure 1 CXCL10 mRNA expression and IgG anti-spike antibody titers stratified by sex, vaccine scheme, time since last dose, and vaccine types.

Boxplots show the log2 fold change in CXCL10 mRNA expression in response to Wuhan and Omicron peptide stimulation (top and middle panels), and log10-transformed anti-S1 IgG antibody titers (bottom panels). Comparisons were made according to sex (female vs. male), vaccine scheme (heterologous vs. homologous), time since the last dose (≤6 months vs. >6 months), and administration of BBIBP-CorV, ARNm-1273, or different two-dose regimens (BBIBP-CorV vs. BNT162b2). Statistical differences between groups were evaluated using either the Wilcoxon rank-sum test or the Student’s t-test, as appropriate. p-values are indicated as follows: p < 0.05 (*), p < 0.01 (**); ns = not significant.

Regarding the time since the last vaccination, no significant differences in CXCL10 expression levels were identified. However, participants who had received their last dose within the past 6 months exhibited higher IgG anti-spike antibody levels (Wilcoxon test, p = 0.001).

Concerning the vaccination regimen, no significant differences were found between homologous and heterologous schedules or when grouping participants based on whether the first two doses corresponded to BBIBP-CorV or BNT162b2 vaccines (Fig. 1). Nevertheless, the inclusion of the mRNA-1273 vaccine in the regimen was associated with higher IgG anti-spike antibody levels compared to participants whose regimens did not include this vaccine (Wilcoxon test, p = 0.002). It is important to note that the inclusion of the BNT162b2 (n = 35 vs n = 5) and AZD1222 (n = 6 vs n = 34) vaccines could not be adequately assessed due to significant imbalance in group sizes.

Immunity levels in relation to quantitative clinical variables

The multivariate analysis of −ΔΔCt Omicron, −ΔΔCt Wuhan, and anti-spike IgG values in relation to clinical quantitative variables did not reveal statistically significant coefficients (Table 2). However, when evaluating the individual correlation of these variables with immune response levels, statistically significant associations were observed between anti-spike IgG antibody levels and age (Spearman’s ρ = 0.33, p = 0.038), the time elapsed since the last vaccine dose (Spearman’s ρ = −0.47, p = 0.004), and the number of vaccine doses received (Spearman’s ρ = 0.55, p = 0.0002) as shown in Fig. 2.

Table 2 Multivariate analysis of factors associated with differential CXCL10 expression and anti-spike IgG antibody levels against SARS-CoV-2.

	−ΔΔCt Wuhan	−ΔΔCt Omicron	Log IgG anti-spike	
	β	p-value	β	p-value	β	p-value	
(95% CI)	(95% CI)	(95% CI)	
Age	0.108	0.436	0.203	0.222	0.001	0.735	
[−0.17 to 0.39]		[−013 to 0.54]		[−0.003 to 0.005]		
Number of COVID episodes	−0.025	0.988	1.528	0.459	0.008	0.769	
[−3.56 to 3.51]		[−2.66 to 5.71]		[−0.05 to 0.06]		
Last COVID episode (months)	−0.072	0.613	−0.181	0.284	0.0004	0.857	
[−0.36 to 0.22]		[−0.52 to 0.16]		[−0.004 to 0.005]		
Number of vaccines received	−0.766	0.773	0.454	0.885	0.045	0.253	
[−6.19 to 4.65]		[−5.96 to 6.87]		[−0.03 to 0.12]		
Last vaccine received (months)	−0.169	0.541	0.018	0.956	−0.003	0.420	
[−0.73 to 0.39]		[−0.65 to 0.68]		[−0.012 to 0.005]		
Note:

CI, Confidence interval.

Figure 2 Correlation analysis of IgG anti-Spike levels with quantitative variables.

(A) Age. (B) Number of vaccine doses. (C) Time since last dose. The red lines represent the fitted regression models, and the shaded areas indicate the 95% confidence intervals. R, Spearman correlation coefficient.

Discussion

Numerous studies have demonstrated that immunity acquired against SARS-CoV-2 wanes over time, a phenomenon exacerbated by the emergence of VOCs (Menegale et al., 2023). Vaccination remains the most effective strategy to sustain robust immunity against the virus. In this regard, it is critical to implement new public health policies within vaccination programs during the post-pandemic phase to protect both the general population and high-risk groups. For a comprehensive understanding of SARS-CoV-2 immunity, it is essential to jointly evaluate both humoral immunity and cellular responses, as their complex interplay is key to protection against infection (Schwarz et al., 2022a).

Since the introduction of the first SARS-CoV-2 vaccines, initial efforts focused on immunizing high-risk groups and older adults, later extending to the general population. However, the global demand for vaccines and the emergence of immune-evasive variants posed significant challenges to achieving herd immunity during the early years of the pandemic (Liu et al., 2021).

In the present study, humoral and cellular immunity against SARS-CoV-2 was evaluated in a cohort of 40 Peruvian individuals. Although the sample size is limited and subject to inherent selection bias due to recruitment from a single center, this type of exploratory research remains valuable. It provides preliminary evidence necessary to guide larger-scale studies and helps characterize immunological phenomena in underrepresented populations. Moreover, studies with small sample sizes have proven to be valuable during the pandemic, as their inclusion in evidence integration studies contributes to a robust evidence base without significantly altering overall findings (El Alayli et al., 2023).

In the post-pandemic era, mass vaccination campaigns, supported by greater vaccine availability and the effectiveness of new formulations against VOCs, have considerably improved the prospects of achieving herd immunity, a goal that once seemed unattainable (Aschwanden, 2021). However, the decline in SARS-CoV-2 cases has led to widespread complacency regarding vaccination schedules. This is evident in our study, where only 25% of participants received their last dose within the past 6 months, and less than half had received more than three doses. This trend could undermine population-level protection, leaving room for a new wave of infections should a novel VOC emerge.

Cellular immunity assessment has unveiled critical insights into the durability and dynamics of the immune response against SARS-CoV-2, emphasizing its crucial role in preventing severe forms of the disease (Petrone et al., 2023). While viral variants can reduce the efficacy of neutralizing antibodies, T cell responses remain remarkably robust (Geers et al., 2021). This highlights the necessity of incorporating T cell evaluation for a comprehensive understanding of protective immunity. However, large-scale measurement of these responses remains challenging due to technical limitations and the lack of standardized methodologies (Schwarz et al., 2022b).

In this study, an innovative approach was employed based on the measurement of CXCL10 mRNA expression, a gene upregulated in monocytes by IFN-γ, which is produced upon antigen-specific T cell activation in response to SARS-CoV-2 peptides (Schwarz et al., 2022b). Using this method, cellular immunity to SARS-CoV-2 was detected in a high percentage of participants (97.5% for Omicron and 90% for Wuhan). Similar findings were reported in a population-based study in Italy, which demonstrated 92.1% positivity using the same molecular approach (Cosma et al., 2023). These studies are among the earliest to implement this novel approach for assessing cellular immunity to SARS-CoV-2.

Although the approach proposed by Schwarz et al. (2022b) does not directly quantify the number of antigen-specific T cells, it assesses their functional activity by measuring their ability to produce IFN-γ in response to viral peptides presented by major histocompatibility complex molecules. In this context, CXCL10 mRNA expression levels measured through the 2−∆∆Ct method may qualitatively indicate the magnitude of cellular immunity, analogous to qualitative analyses in IFN-γ release assays (Cosma et al., 2023).

The presence of IgG in the serum of all participants indicates a specific humoral response that could offer some degree of protection against infection. This high positivity rate might be attributed not only to vaccination but also to widespread and often unnoticed exposure to the virus in everyday life (Swadźba et al., 2024). Regarding the duration of the humoral response, previous studies have shown that IgG levels decline approximately 6 months after vaccination (Đaković Rode et al., 2022; Karl et al., 2023). This pattern aligns with and is reinforced by the negative correlation observed in this study between the time since the last vaccine dose and IgG levels.

However, some studies have reported that IgG levels can remain relatively stable for up to 1 year (Guo et al., 2024), and seropositivity may persist for at least 20 months (Dobaño et al., 2022). This could explain the presence of anti-spike IgG antibodies in participants vaccinated more than 6 months ago. Nevertheless, IgG levels do not always correlate with long-term protection against reinfection, particularly in the context of emerging VOC (Franco-Luiz et al., 2023). Protection may improve with an increased number of vaccine doses (Matsumoto et al., 2024), as also observed in this study, where correlation analyses showed a positive relationship between IgG levels and the number of doses received.

On the other hand, previous studies on cellular immunity in SARS-CoV-2-vaccinated populations have found no significant correlation between immune response and sex (Graça et al., 2023; Misra et al., 2023). Similarly, other studies have not identified differences in IgG levels between men and women (Marchevsky et al., 2022; Trevisan et al., 2023). However, some studies have reported that women tend to develop a stronger immune response following vaccination (Fernandes et al., 2023). In contrast, this study found that male participants exhibited significantly higher immune responses than female participants. This trend, though mild, has also been observed in other studies, particularly in populations vaccinated with mRNA-based platforms (Korosec et al., 2024). Additionally, these results may be influenced by the higher percentage efficacy of certain vaccines in men (Jensen et al., 2022). Collectively, these findings suggest the presence of contextual or population-specific factors that may influence immune responses based on sex, which remain to be fully elucidated.

Studies in vaccinated populations have demonstrated that age does not significantly influence cellular immune positivity, as observed in groups from France (Graça et al., 2023) and India (Misra et al., 2023), findings consistent with those obtained in the Peruvian population studied here. However, other investigations have reported a decline in cellular immunity among individuals over 75 years old (Dietz et al., 2023). Regarding humoral immunity, a negative correlation between age and immune response following vaccination has been reported (Fernandes et al., 2023; Hasan et al., 2023). In contrast, this study found a positive correlation between age and IgG levels, agreeing with findings in healthcare workers 2 years after their first dose (Swadźba et al., 2024). These results suggest that the influence of age on the immune response may vary depending on the population context and study design.

No significant differences in humoral and cellular immunity were observed between homologous and heterologous vaccination schemes in this study. However, previous research has shown that heterologous vaccination provides more durable protection against SARS-CoV-2 compared to homologous strategies (Costa Clemens et al., 2022; Orlandi et al., 2023; Adnan et al., 2024). This evidence has led many countries to adopt heterologous vaccination strategies in their immunization programs (Siddiqui et al., 2022). The advantage of heterologous schemes has already been reported in Peruvian healthcare workers, who demonstrated enhanced humoral immune responses; however, it was also noted that the time elapsed since the booster dose was associated with lower geometric mean IgG levels (Montero et al., 2023). Additionally, it is important to note that the immune response following vaccination tends to stabilize over time (Makri et al., 2024). These factors may explain the lack of observed differences in immune responses between vaccination schemes in this study.

To date, none of the various SARS-CoV-2 vaccines have shown to be significantly superior in terms of efficacy (Wu et al., 2024). However, it has been observed that vaccination schedules combining different types of vaccines may result in variations in cellular and humoral immunity (González et al., 2023). When grouping the population based on the vaccines included in their schedules, such as the inclusion of BBIBP-CorV and BNT162b2 vaccines in the first two doses, which were the most prevalent and available from the beginning of the vaccination program in Peru (López et al., 2022), no significant differences in cellular and humoral responses were found. However, when grouping the population according to the inclusion of certain vaccines, higher IgG levels were found in individuals who received the mRNA-1273 vaccine. This finding could be explained by the tendency of mRNA-based vaccines to show higher efficacy (Salinas-Martínez et al., 2023; Wu et al., 2024). Nevertheless, further studies are needed, as these findings may be influenced by environmental factors and the genetic diversity of the Peruvian population (Van Dorst et al., 2024).

Overall, previous research in various populations has shown that cellular immunity is more durable and remains functional against VOCs (Almendro-Vázquez, Laguna-Goya & Paz-Artal, 2023; Nesamari et al., 2024). Additionally, it has been reported that inactivated virus vaccines maintain stable activation of CD4+ T lymphocytes, both for the ancestral SARS-CoV-2 strain and the Omicron variant over time (Méndez et al., 2023). Similarly, it has been observed that the reactivity of CD4+ and CD8+ T cells does not vary significantly in response to variants in mRNA vaccine receptors (Tarke et al., 2021). These findings may explain the high cellular response to the Omicron variant observed in the studied population, as most individuals received vaccination schedules based on inactivated virus (BBIBP-CorV) and mRNA (BNT162b2) vaccines. However, this does not explain the higher cellular response to Omicron compared to the original Wuhan strain. Furthermore, this contrasts with reports from other studies where the response to the original Wuhan strain was greater than to the Omicron variant (Keeton et al., 2022). The superior response to Omicron could be related to the widespread exposure of the Peruvian population to this variant during previous infection waves, during which Omicron was responsible for a significant increase in cases (Dámaso-Mata, 2022; Swadźba et al., 2024).

This study presents a preliminary analysis of the immune response in the Peruvian population, employing a recently developed methodology. Limitations of this study include the small sample size and the inherent selection bias resulting from recruitment at a single center, which restrict the generalizability of the findings. Therefore, the interpretations presented in the discussion should be regarded as initial approximations that may be strengthened through future studies with broader representativeness. Additionally, the study focused exclusively on the cellular response to the Omicron variant, without including other variants of interest. Expanding such analyses to include additional variants could provide a more comprehensive understanding of the immune landscape in this population.

Conclusions

Our results showed that most Peruvian participants exhibited robust immunity, with IgG antibodies present in all individuals and a high proportion showing a positive cellular response, especially to the Omicron variant. While no significant differences were observed between vaccination scheme, the inclusion of the mRNA-1273 vaccine was associated with higher IgG levels. Additionally, IgG levels showed a moderate correlation with the number of doses received and a weak inverse correlation with age, with higher levels observed in participants who had recently received their latest dose. Despite these findings, the trend towards a decrease in vaccination coverage raises concern as it could weaken population immunity and increase the risk of reinfections, particularly with the emergence of new variants. These results highlight the importance of continued surveillance and the need to adjust vaccination strategies to maintain adequate protection and public health resilience.

Supplemental Information

Supplemental Information 1 Characteristics of study participants, IgG values and CXCL10 relative expression levels.

Supplemental Information 2 MIQE checklist.

Supplemental Information 3 STROBE checklist.

We gratefully acknowledge Lorena Barrios, Rosario Oporto and Daniella Orihuela for their valuable help during sample collection and processing.

Additional Information and Declarations

Competing Interests

The authors declare that they have no competing interests.

Author Contributions

Marlon Yuri Garcia-Paitan conceived and designed the experiments, performed the experiments, analyzed the data, prepared figures and/or tables, authored or reviewed drafts of the article, and approved the final draft.

Salyoc Tapia-Rojas analyzed the data, prepared figures and/or tables, authored or reviewed drafts of the article, and approved the final draft.

Hector Santiago Alvarez Vega performed the experiments, authored or reviewed drafts of the article, and approved the final draft.

Javier Enciso-Benavides performed the experiments, authored or reviewed drafts of the article, and approved the final draft.

Maria J. Pons conceived and designed the experiments, authored or reviewed drafts of the article, and approved the final draft.

Ana Mayanga-Herrera conceived and designed the experiments, analyzed the data, authored or reviewed drafts of the article, and approved the final draft.

Human Ethics

The following information was supplied relating to ethical approvals (i.e., approving body and any reference numbers):

The Institutional Review Board of the Universidad Científica del Sur (CIEI-CIENTIFICA) approved this study under approval number 173-CIEI-CIENTIFICA-2023.

Data Availability

The following information was supplied regarding data availability:

The dataset used for statistical analysis is available in the Supplemental File.

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
