# Peer review of "SARS-CoV-2-specific humoral and cellular immunity assessment in Peruvian vaccinated population: a cross-sectional study"

_PeerJ, doi:10.7717/peerj.19651_

## Round 0.1 · original submission · Major Revisions

· Academic Editor

Major Revisions

The authors are required to provide major revision of the manuscript. The focus should be on experimental design and validity of the findings.

Reviewer 1 ·

Basic reporting

no comment

Experimental design

no comment

Validity of the findings

no comment

Additional comments

The article is well-written and clear. The methods are straightforward and the results are shown appropriately.

Reviewer 2 ·

Basic reporting

The title is clear, presenting the aim of the study and the study finding. The manuscript is well-written, and clear to understand for the rational background of the study. Although the study finding did not show the positive finding between observed group, it is still understandable and the reason for its phenomenon is rational with the result given (in figure, table or graph), as presented in the result section and discussion.

Experimental design

The design is clear, no specific comment arise for this.

Validity of the findings

The main limitation of this study is the study population. The study population is small (only 40 subject included) and single health facility center. Author has tried the best to describe the phenomenon occurred in this 40 subject, however, it is need to describe as the limitation of the study. Such as selection bias of population seems can not be avoided, and many extrapolated argumentation was made in the discussion. Suggestion, add limitation of the study of study population as the main interest of discussion.

·

Basic reporting

The manuscript showed that most Peruvian participants exhibited robust immunity, with IgG
antibodies present in all individuals and a high proportion showing a positive T-cell immunity response,
determined by RT-qPCR analysis of CXCL10 mRNA as a surrogate for IFN-gamma.

Experimental design

However, the Methods were not described with sufficient detail and information to replicate. The following points need to be addressed:
1. The authors need to provide more detailed descriptions of their methods. For the RT-qPCR, please specify the ratio of OD260 / OD280 in the isolated RNA. The authors employed a very primitive method without column purification; please clarify whether DNase A was used to digest the genomic DNA. Additionally, indicate whether the CXCL10 primers were designed to target exon-exon junctions. Lastly, please confirm if the negative control, which did not include the reverse transcription step, also amplified genomic DNA.
2. For Fig. 1, in y-axis, is it 2^Delta? There is no 2^-DeltaDelta. Did the authors normalize the results with the control group CXCL10?
3. For Fig. 1 and 2 IgG anti-Spike antibody titer, the Log10 value seems to be around 0.7 to 1.2. The cut-off in the method indicated a ratio >1.1 for positivity. What are the titers? The IgG antibody analysis method should be described in more detail. Readers will not have time to check the reference for the method. Please consider the reference line 414, Cosma et al, as an example.

Validity of the findings

The manuscript showed that most Peruvian participants exhibited robust immunity, with IgG
antibodies present in all individuals and a high proportion showing a positive T-cell immunity response,
determined by RT-qPCR analysis of CXCL10 mRNA as a surrogate for IFN-gamma. Conclusions are well stated and linked to the original research question.

---

## Round 0.2 · accepted · Accept

· Academic Editor

Accept

The authors addressed all comments and manuscript can be accepted.

·

Basic reporting

The manuscript has assessed SARS-CoV-2-specific humoral and cellular immunity in Peruvian vaccinated population. Sufficient background is provided The figures and tables are clear, and the raw data are shared.

Experimental design

Research question is well defined, relevant & meaningful. It is stated how research fills an identified knowledge gap. Methods are described with sufficient detail & information to replicate.

Validity of the findings

All underlying data have been provided; they are robust, statistically sound, & controlled. Conclusions are well stated, linked to original research question & limited to supporting results.

Additional comments

The authors have addressed my comments.